# SafeMPO: Constrained Reinforcement Learning with Probabilistic Incremental Improvement

**Alexander Mattick[1], Dominik Seuß[2,1], Christopher Mutschler[3,1]**
[1] Fraunhofer Institute of Integrated Circuits IIS, Fraunhofer IIS, Nürnberg, Germany
[2] Technical University of Applied Sciences Würzburg-Schweinfurt, Germany
[3] University of Technology Nuremberg (UTN), Germany
{firstname.lastname}@iis.fraunhofer.de

## Abstract

Reinforcement Learning (RL) has demonstrated significant success in optimizing complex control and planning problems. However, scaling RL to real-world applications with multiple, potentially conflicting requirements requires an effective handling of constraints. We propose a novel approach to constraint satisfaction in RL algorithms, focusing on incrementally improving policy safety rather than directly projecting the policy onto a feasible region. We accomplish this by first solving a nonparametric surrogate problem which is guaranteed to contract towards the feasible set, and then cloning that solution into a neural network policy. As a result, our approach improves stability, particularly during early training stages, when the policy lacks knowledge of constraint boundaries. We provide general theoretical results guaranteeing convergence to the safe set for this class of incremental systems. Notably, even the simplest algorithm produced by our theory produces comparable or superior performance when compared to highly tuned constrained RL baselines in challenging constrained environments.

## 1 Introduction

Reinforcement Learning (RL) offers significant potential for improving performance in a wide range of control and planning problems by directly optimizing the behavior policy through interactions with a black-box environment (Sutton & Barto, 2018). However, RL can pose safety challenges due to an untrained policy that accidentally damages equipment or even harms users during the learning process. Further, even if a policy has been safely trained to completion (e.g. in a simulator), one can generally not be sure whether the policy can be considered safe during deployment in the real world.

To address these issues, there is a growing trend towards Safe Reinforcement Learning (SRL) which explicitly introduces safety considerations into the training process. One of the most popular formulations for SRL is the Constrained Markov Decision Process (CMDP), which extends the traditional Markov Decision Process (MDP) by incorporating constraints $C(a, s)$ on the policy's behavior:

$$\max_{\pi} R(\pi) \quad \text{s.t.} \quad C_i(\pi) \leq B_i \quad \forall i = 0, \dots, N, \tag{1}$$

where $R(\pi) = \mathbb{E}_{\tau \sim \pi}[\sum_{t=0}^{\infty} \gamma_r^t r_t]$ represents the expected discounted future rewards, and $C_i(\pi) = \mathbb{E}_{\tau \sim \pi}[\sum_{t=0}^{\infty} \gamma_c^t c_{t,i}]$ the expected discounted future costs for the $i$-th cost function. For the sake of this work, we define $C(s, a)$ as the cost associated with executing action $a$ in state $s$ and following the policy after (analogous to the Q-function). *Constrained Reinforcement Learning (CRL)* refers to the process of finding a policy that is optimal and is within the feasible set defined by Eq. 1.

It is worth noting that CRL is of wider interest than just safety. Constraints can be used to encode a wide range of different requirements, such as ensuring an autonomous quadcopter has enough battery to return to its charging station, or preventing policies from abusing known inaccuracies in a simulation of the real-world environment. Indeed, many reward shaping problems can be efficiently reformulated as constraint satisfaction tasks.

Methods that optimize directly for safety can struggle when the set of feasible policies is difficult to find or when the *closest* feasible policy is very suboptimal. Fig. 1 shows a robot that attempts

to throw a ball onto one of three platforms, each of them separated by chasms. The initial (unsafe) policy throws the ball onto the middle platform (R:0, C:0) and potentially into the holes to the left (R:1, C:1) and right (R:0, C:1). Since the initial policy has never reached the left platform, a greedy projection of the policy onto the feasible set would mean the policy only hits the central column, terminating with a reward of 0 and a cost of 0. Policy optimization methods that rely on greedy recovery (Achiam et al., 2017; Xu et al., 2020) or projection (Narasimhan, 2020; Yang et al., 2022) will naturally find themselves in a local optimum that is safe, but do not obtain reward (or, more generally, an arbitrarily bad reward), due to a lack of exploration.

Such situations are common in practice as they may also arise from inherent randomness in sampling from stochastic environments with stochastic policies. A sufficiently small probability of reaching high-reward, safe areas can effectively eliminate potentially feasible policies from consideration. Ensuring a sufficiently close policy $\pi_{k+1}$ using KL-ball constraints $KL(\pi_{k+1}\|\pi_k) < \varepsilon$, introduces a new challenge, as the intersection of feasible policies and policies within the $\varepsilon$-ball might be empty.

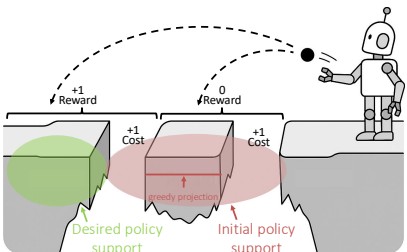

This is a known problem: Constrained Variational Policy Optimization (CVPO) (Liu et al., 2022) aims to directly solve a nonparametric surrogate problem. However, there is no guarantee that a safe policy within a KL-ball exists as the initial policy can be arbitrarily far away from the feasible set. As a result, Liu et al. (2022) decay their cost upper bound $B$ from some large value $B_{max} \to B$ over

Figure 1: A robot throws a ball to a different platform. The initial policy (gray) either hits the central platform (R:0) or one of the two chasms (C:1). The optimal policy throws the ball directly to the platform on the left (R:1, C:0).

the course of training. The limitation of this is obvious: Too fast decay may lead to unsatisfiable problems, while too slow decay requires a large number of iterations exceeding $B$. This means that the decay speed is a crucial hyperparameter that is also difficult to tune in practice, since it largely depends on the exploration performance and the specifics of the constraints and the environment.

One might challenge this notion with the proof provided by Paternain et al. (2019) who show that constrained RL has zero duality gap.[1] Indeed, this is theoretically true: Assuming the occupation measure has a nonzero probability to reach every state, one could discover every possible policy. However, this relies on the assumption that the occupation measure has a nonzero probability of reaching every state, which is unrealistic in practical algorithms with finite evaluations. Low-probability states effectively disappear from the sampled distribution and are never optimized. Since RL over occupation measures is linear (see (Paternain et al., 2019), program $PI''$), the distribution will never get optimized increased occupation in those states.

We propose an alternative approach: instead of greedily projecting onto the safe set, we aim to solely ensure that policy $\pi_{k+1}$ is *safer* than the previous policy $\pi_k$. We formulate this as a constrained Bayesian optimization problem, seeking a *monotonically increasing* likelihood of a trajectory being safe $p(S = 1)$. Being a Bayesian approach this is typically more robust to approximation errors and local optima. This way we can design algorithms that achieve high safety, while being more robust against approximation errors and local optima.

## 2 RELATED WORK

Contemporary methods for achieving constraint satisfaction on CMDPs often rely on identifying and maintaining safety once a safe policy has been found. For instance, Constrained Policy Optimization (CPO) (Achiam et al., 2017) bounds the increase in costs ensuring that given a safe policy $\pi$ the updated policy $\pi'$ will remain safe. During training, whenever a policy is considered unsafe, cost-minimizing recovery steps are performed. Constrained Update Projection (CUP) (Yang et al., 2022) utilizes a similar bound to project the updated policy back onto the feasible set defined by a generalization of the inequality used in (Achiam et al., 2017). Constraint-Rectified Policy

---

[1]Indeed the proof by Paternain et al. (Paternain et al., 2019) rediscovered a more general notion: every optimization problem $\max_x L(x)$ can be turned into a linear program over measures: $\max_\rho \int L(x)\rho(x)dx$, with the minimum being obtained by e.g. the Dirac impulse at the optimal $x^\star$ (see e.g., (Lasserre, 2009)).

Optimization (CRPO) (Xu et al., 2020) updates the policy towards the optimum as long as the policy is safe, reverting to cost minimization to regain safety otherwise. Reward Constrained Policy Optimization (RCPO) (Tessler et al., 2018) combines the reward and cost signal into a single value function $V(s_t, \lambda)$ parameterized by the dual $\lambda$ allowing for more principled updates than a traditional Lagrangian approach. This method is limited by how quickly $\lambda$ can be updated under exploration noise. Projection-Based Constrained Policy Optimization (PCPO) (Narasimhan, 2020) directly projects the policy onto the feasible set in every iteration, but requires Hessian calculations to perform accurate projections. Furthermore, projections may lead to large jumps in policies especially during early training as the estimate of the feasible set may be very noisy and the policy may be far from feasible. Stooke et al. (2020) adopt a PID perspective for tuning the Lagrangian $\lambda$ which gives rise to a host of different traditional methods combined with a penalty function. Constraint recovery steps (either via hard projection as in CUP or descent-based as in CPO) easily lead to the scenario from Fig. 1. This is because the projection onto the space of safe distributions may not be exact (e.g., due to sampling errors) which prematurely reduces exploration for states that may be safe in practice. Our method solves this by not overcommitting to any individual cost estimate.

Outside of CRL, one notable earlier work is Maximum a Posteriori Optimization (MPO) (Abdolmaleki et al., 2018). MPO first optimizes a surrogate objective using a nonparametric approximation $q$ of the policy-improvement objective. The result of the surrogate is then cloned to the policy neural network by minimizing $KL(\pi\|q)$. This explicit surrogate means that, from the point of view of neural network optimization, reinforcement learning becomes a supervised problem of matching to a known target distribution.

RL has long relied on incremental objectives, such as advantage functions, to reduce variance and improve convergence. It stands to reason that a similar treatment would benefit constrained RL methods. Indeed, most convergence proofs used in reinforcement learning rely on some notion of relative improvement compared to the previous policy (e.g. (Sutton & Barto, 2018; Xiao, 2022)). We would argue that such guaranteed improvement constraints are more natural in the iterative "explore then optimize" framework commonly used in RL.

## 3 SAFE MAXIMUM A POSTERIORI OPTIMIZATION (SAFEMPO)

We build upon Maximum a Posteriori Optimization (MPO) (Abdolmaleki et al., 2018), extending their EM-style RL update to CMDPs. MPO sees policy updates as inference problems with the assumption that the probability of a trajectory being optimal $p(O = 1)$ follows a Boltzman distribution $p(O = 1|\tau) \propto \exp(\sum_t r_t/\alpha)$. This leads to the following ELBO

$$\log P_\pi(O = 1) = \log \int p_\pi(\tau)p(O = 1|\tau)d\tau \geq \int q(\tau)\left[\log p(O = 1|\tau) + \log\frac{p_\pi(\tau)}{q(\tau)}\right]d\tau$$

$$= \mathbb{E}_q\left[\sum_t r_t/\alpha\right] - KL(q(\tau)\|p_\pi(\tau)), \tag{2}$$

which is then solved in two phases. This is a traditional Bayesian inference problem which can be solved using the Expectation-Maximization (EM) algorithm. The EM algorithm (Neal & Hinton, 1998) solves Equation 2 by iteratively maximizing $q$ or $\pi$ while keeping the other fixed. In the case of MPO, we first solve a nonparametric approximation derived directly from a batch of samples from the replay buffer to obtain $q$:

$$\max_q \ \mathbb{E}_{(a,s)\sim q}[Q(a,s)] \tag{3a}$$

$$\text{s.t.} \ \ KL(q\|\pi) \leq \varepsilon \tag{3b}$$

$$\int q(a,s)dads = 1 \tag{3c}$$

where the KL-constraint is introduced to automatically construct a suitable temperature $\alpha$. In a second step, one matches the policy neural network $\pi$ to the nonparametrically optimal $q$

$$\min_\pi\{KL(q\|\pi) \ : \ KL(\pi_{\text{old}}\|\pi) < \varepsilon\}. \tag{4}$$

This step is not dependent on any potentially nonstationary objective, but only has to clone the information of the optimal distribution found in eq. (3).

MPO now proceeds to alternatingly solve Equation (3) and Equation (4): In every iteration, the current policy $\pi$ is approximated empirically by drawing states from the replay buffer and predicting $\pi(a|s)$. The samples are used to solve Equation (3) for the optimal $q^2$, which is then cloned into the policy network using Equation (4).

We believe this to be an appealing structure for CRL as the nonparametric model eq. (3) can easily accomodate additional constraints, which can then be handled by efficient optimizers (in our case Sequential Least Squares Programming).

## 3.1 SAFEMPO

We generalize this to CMDPs by introducing a second "policy is safe" event $S$ and then optimizing the resulting joint distribution:

$$\log P_\pi(O = 1, S = 1) = \log \int p_\pi(\tau) p(O = 1|\tau) p(S = 1|\tau) d\tau \tag{5a}$$

$$\geq \int q(\tau) \left[ \log p(O = 1|\tau) + \log p(S = 1|\tau) + \log \frac{p_\pi(\tau)}{q(\tau)} \right] d\tau \tag{5b}$$

$$= \mathbb{E}_q \left[ \sum_t r_t/\alpha \right] + \log p(S = 1|\tau) - KL(q(\tau)\|p_\pi(\tau)) \tag{5c}$$

Following Abdolmaleki et al. (2018), we define the probability of safety as an exponential law, but with the notable change that we truncate the law while the policy is safe

$$p(S = 1|\tau) \propto \begin{cases} \exp\left( \frac{-(C(a,s)-B)}{\beta} \right) & C(a,s) \geq B \\ 1 & \text{otherwise} \end{cases} \tag{6}$$

Notice that this truncation automatically makes sure that all feasible policies have an equal likelihood to be safe, which is exactly the behavior we want in CRL. We also define the cost-log-likelihood function:

$$G(a,s) = \log p(S = 1|\tau) = -\max\left( \frac{C(a,s)-B}{\beta}, 0 \right). \tag{7}$$

This is the canonical choice for mapping the costs to a distribution, just as $\exp(R(\tau))$ is the canonical choice for returns (Abdolmaleki et al., 2018). Specifically, the Gibbs distribution is the unique distribution that weights higher $R$ higher while keeping maximal entropy (Jaynes, 1957). Our truncation adds the third property that values above $B$ are considered equal, which is the core difference between an optimization objective, and a constraint.

Notice that Eq. 5 allows for a wide variety of optima: Any weighting for safety against optimality is possible. As we are interested in CRL (instead of multi-objective RL), we try to select an optimum such that the new policy $q$ is *at least* as safe as the old policy $\pi$. Like Abdolmaleki et al. (2018), we replace the temperature $\beta$ with a constraint, just that we want to use an improvement bound **safety**$(\pi) \leq$ **safety**$(q)$.

**Remark 1.** *Note that our proofs would work, even if we choose a different distribution/likelihood function $G$ since our proofs only rely on a safer policy having a higher likelihood. Therefore, any function that is monotonically decreasing for values $C_i(\pi) \leq B_i$ will provide a feasible safety likelihood. We choose $G$ as a truncated exponential, due to its connection with the reward function used in (Abdolmaleki et al., 2018; Haarnoja et al., 2017). However, this choice has disadvantages, which we will discuss in Section 4.*

In the following, we will show that with very limited assumptions, a simple improvement constraint **safety**$(\pi) \leq$ **safety**$(q)$ is *insufficient* to guarantee improvement, and how one can create a consistently improving policy by explicitly forcing improvements in every iteration. First, to formalize the notion of **safety**$(\pi) \leq$ **safety**$(q)$, we introduce the notion of a safety order preservation function:

---

[2]In practice, one solves the dual problem which is equivalent due to Equation (3) being a convex problem with nonempty interior

**Definition 1** (Safety Order Preserving Functions). *A function $K(p, q)$ is called safety order preserving if, for all distributions $q, \pi$ and all functions $G(a, s)$ we have*

*1.* $\mathbb{E}_{(a,s)\sim q}[G(a,s)] > \mathbb{E}_{(a,s)\sim\pi}[G(a,s)] \implies K(q,\pi) > 0,$

*2.* $\mathbb{E}_{(a,s)\sim q}[G(a,s)] < \mathbb{E}_{(a,s)\sim\pi}[G(a,s)] \implies K(q,\pi) < 0$

*3.* $\mathbb{E}_{(a,s)\sim q}[G(a,s)] = \mathbb{E}_{(a,s)\sim\pi}[G(a,s)] \implies K(q,\pi) = 0$

Informally, a Safety Order Preserving function is one that defines a "direction" towards the feasible set. This definition will allow us later to make general monotonic improvement guarantees by defining "improvement" in terms of a general $K(\cdot, \cdot)$ function, which may allow individual states to increase cost, while decreasing the cost over all, facilitating greater freedom of exploration. We will now give examples for Safety Order Preserving Functions, starting with the naive choice of a linear improvement in expectation:

**Example 3.1** (Linear Safety Function (LSF)).

$$K(q, \pi) = \int q(a, s)G(a, s)dads - \int \pi(a, s)G(a, s)dads$$

*is safety order preserving.*

**Example 3.2.** *Priors conditional on safety*

$$K(q, \pi) = \begin{cases} f(q) & \int q(a,s)G(a,s)dads > 0 \\ 0 & \int q(a,s)G(a,s)dads = 0 \\ g(q) & otherwise \end{cases}$$

*for $f$ is a positive, $g$ is a negative function, are safety order preserving.*

Flexibility in the safety order preserving function allows us to e.g. favor safe policies with lower variance, or to favor policies deeper in the interior of the feasible set by choosing a suitable $f$. It also allows a "hard" improvement boundary which simply counts the number of safe trajectories by setting $f = 1$ and $g = -1$. We will keep a more thorough analysis of good choices for $K$ for future work. For now, the framework of safety order preserving functions allows us to make very general statements about how incremental improvement constraints can be used in CRL.

**Theorem 1.** *Let $Q(a, s)$ be a Q-function, $G(a, s)$ the cost-likelihood function, $\pi$ the current policy, and $K(p, q)$ safety order preserving function. Then the optimal nonparametric policy $q$ solving the improvement equation*

$$\max_q \; \mathbb{E}_{(a,s)\sim q}[Q(a,s)] \tag{8a}$$

$$\text{s.t.} \quad KL(q\|\pi) \leq \varepsilon \tag{8b}$$

$$K(q, \pi) \geq 0 \tag{8c}$$

$$\int q(a,s)dads = 1 \tag{8d}$$

*either has the same safety as the original $\pi$, or ignores $K$ and solves the unconstrained MPO problem (Eq. 3).*

Proofs can be found in Appendix B. This theorem immediately gives rise to two corollaries:

**Corollary 1.** *Whether a policy ends up being safe under the incremental model **completely depends** on whether $Q(a, s)$ is higher in safe regions than in unsafe ones.*

**Corollary 2.** *Choosing the naive improvement constraint*

$$\mathbb{E}_{(a,s)\sim q}[G(a,s)] \geq \mathbb{E}_{(a,s)\sim\pi}[G(a,s)]$$

*for Equation (8) is **not** a contraction over the safety likelihood $G$.*

### 3.2 GUARANTEED IMPROVEMENT E-STEP

The core issue behind the method outlined in Theorem 1 is that they do not force the model to pick a point in the interior of the feasible set of Eq. 8. This means we cannot guarantee a non-zero improvement in safety. To solve this issue, we borrow techniques from interior point methods, namely logarithmic barrier functions.

In interior point methods, the log-barrier function is used to maintain constraints $g(x) \leq 0$ by converting them to a barrier of the form $\mu \log(-g(x))$. This ensures that once a point is feasible, it will always stay feasible due to the singularity at $\log(0)$. The downside for classical optimization is that points close to the barrier $g(x) = 0$ are hard to reach, which is why barrier methods slowly *reduce* the barrier weight $\mu \to 0$ during optimization to approximate the true feasible region. This way interior point methods slowly walk closer towards the barrier, but can never jump over $g(x) = 0$ due to the singularity of the logarithm.

Luckily, as we expressly *want* points in the interior, we can pick any arbitrary $\mu$ and a logarithmic barrier to ensure a nonzero improvement in our safety constraints. Specifically, with $K$ being a safety-order preserving function, we can formulate our incremental approach as

$$\max_{q} \ \mathbb{E}_{(a,s)\sim q}[Q(a,s)] + \kappa \log\left(\frac{x}{\kappa}\right) \tag{9a}$$

$$\text{s.t.} \ \ KL(q\|\pi) \leq \varepsilon \tag{9b}$$

$$K(q,\pi) \geq x \tag{9c}$$

$$\int q(a,s)dads = 1 \tag{9d}$$

Assuming $\mathbb{E}[Q(a,s)]$ is bounded, this method is guaranteed to improve the safety margin in every iteration by construction and provides a geometric convergence towards the feasible set (see theorem 4). For any $\kappa > 0$, every $q$ with a strictly positive improvement $x > 0$ will be favored over all other $q$ since for $x \leq 0$ we have $\log(x) = -\infty$.[3] Importantly, neither the improvement nor the final optimum when this is iterated depends on the choice of $\kappa$, only the local per-iteration tradeoff between reward maximization and constraint improvement changes as $\kappa$ changes!

For example, when choosing $K$ as the LSF in Example 3.1, $\kappa \to \infty$ produces a hard projection onto $\{q|KL(q\|\pi) \leq \varepsilon, C_i(q) \leq B_i\}$, and for $\kappa \to 0$ one obtains the MPO update (see also Theorem 3). We simply pick $\kappa = 10$ for all of our experiments. This value is not tuned for our experiments, which corresponds to the real-world usecase where the optimal (or even a good) $\kappa$ may not be known.

One can easily solve Eq. 9 by optimizing over the dual variables $\lambda, \nu$ in

$$\mathcal{L}(\lambda,\nu) = \int q^\star(a,s)Q(a,s)dads - \kappa \log \lambda \tag{10a}$$

$$+ \lambda \left( \int q^\star(a,s)G(a,s)dads - \int \pi(a,s)G(a,s)dads \right) \tag{10b}$$

$$+ \nu \left( \varepsilon - \int q^\star(a,s) \log\left(\frac{q^\star(a,s)}{\pi(a,s)}\right) dads \right), \tag{10c}$$

where the optimal $q^\star$ can be computed in closed form as

$$q^\star(a,s) = \frac{1}{Z}\pi(a,s)\exp\left(\frac{Q(a,s) + \lambda G(a,s)}{\nu}\right). \tag{11}$$

Note that the optimization problem (Eq. 10) is convex and only has 2 variables.[4] To guarantee a solution to Eq. 10 corresponds to solutions of Eq. 9, one needs a constraint qualification and strong duality. In general, solving the dual problem can be seen as favorable for constrained RL, as the dual problem's complexity scales with the number of constraints, rather than the number of state-action pairs in the batch. The latter might be very large as one may need a lot of samples to accurately track the expected values in Eq. 9. Convex optimization commonly uses Slater's condition:

---

[3]We make the usual assumption in convex optimization that undefined values receive a $-\infty$ penalty Boyd, Stephen & Vandenberghe, Lieven (2004).

[4]More generally, for a problem with $N$ constraints, we have $1 + N$ variables.

**Proposition 1** (Slater's condition). *If a convex program with constraints $f_0, \ldots, f_n \leq 0$ has a point in the relative interior $f_0, \ldots, f_n < 0$, strong duality holds*

Slater's condition acts as both a strong duality certificate, and a constraint qualification. Verifying Slater's condition can be somewhat tricky for very general constraint sets such as the ones used for Eq. 10. Fortunately, for Eq. 9 we can give an easier way to check condition for strong duality

**Theorem 2.** *Let $(\mathcal{A} \times \mathcal{S}, \Sigma, \mu)$ be a measure space. If $G(a, s) : \mathcal{A} \times \mathcal{S} \to \mathcal{R}$ is not almost everywhere constant, $\pi$ is fully supported and $\varepsilon > 0$, and $K$ is concave, then Eq. 9 has a solution and strong duality holds.*

The proof can be found in Appendix B. Using this theorem we can conclude the following:

**Corollary 3.** *Let $O : \pi \to q$ be an improvement operator induced by Equation (9). Then $O$ is a contraction wrt the pseudo-metric $K(\cdot, \cdot)$.*

**Remark 2.** *Theorem 2 still holds for multiple constraints $G_1, \ldots, G_N$ if one replaces the condition " $G(a, s)$ is not almost everywhere constant" with "there exists a subset $A' \times S' \subseteq A \times S$ with $\mu(A' \times S') > 0$ where none of the $G_i$ are constant."*

This means that, in practice, one only has to ensure that $G(a, s)$ is not constant, and that $\pi$ has nonzero probability for all outputs (which is a standard assumption in both inference and RL), and that whatever $K$ is chosen has to be concave and safety order preserving. For simplicity, we will choose $K$ as the LSF (Example 3.1), since, without further assumptions, this can be seen as the canonical choice for $K$. Choosing a different $K$ might further improve performance by e.g. favoring high-entropy solutions within the safer set. Choosing $K$ choice has an interesting consequence: even if $G(a, s)$ is constant, we can define a reasonable notion of the optimal solution in Eq. 9.

**Theorem 3.** *Assume a sufficiently large upper bound $M > 0$ on the dual variable associated with the improvement constraint. If $K$ is the LSF (see Example 3.1), then solving Eq. 9 with a constrained dual produces a solution to Eq. 9 if $G$ is not constant, and a solution of Eq. 3 if $G$ is constant.*

The result in Theorem 3 provides a very natural way of implementing the nonparametric surrogate: We optimize the dual variables under some sufficiently large constraints (we choose $\lambda \in [-10^6, 10^6]$). If the batch we are optimizing over is constant (e.g., all $(a, s)$ are already safe, random sampling), we automatically make an MPO step, while in all other cases we perform a safe improvement step. We present the complete method in Algorithm 1.

Combining all previous theorems, we obtain a very general convergence guarantee

**Theorem 4.** *Assuming $\mathbb{E}_{(a,s) \sim q}[Q(a, s)]$ is bounded in $[-M, M]$, and the feasible set is not empty, eq. (9) converges towards the feasible set, and $K$ is sequentially continuous on the nonnegative component (cases 1., 3. for definition 1) and not constant. If we additionally choose $K$ as the LSF (example 3.1), then we obtain geometric convergence speed.*

### 3.3 M-STEP

After the optimal nonparametric $q^\star$ is computed, the resulting distribution has to be "cloned" into the policy neural network. Just like MPO (Abdolmaleki et al., 2018), we factor $\pi(a, s) = \pi(a|s)\mu(s)$ and $q(a, s) = q(a|s)\mu(s)$ where $\mu(s)$ is the stationary distribution estimated using the replay buffer. As in MPO, we use a split KL-difference constraint as the prior in our M-step. Therefore our M-step is

$$\mathcal{L}(q, \alpha_1, \alpha_2) = \mathbb{E}_{\mu(s)}\left[\mathbb{E}_{q^\star}\left[\log \pi(a|s)\right]\right] + \alpha_1(\varepsilon_\mu - C_\mu) + \alpha_2(\varepsilon_\Sigma - C_\Sigma), \tag{12}$$

where $C_\Sigma = \mathbb{E}_{\mu(s)}[\frac{1}{2}(\Sigma^{-1}\Sigma_{old}) - n + \log\left(\frac{|\Sigma|}{|\Sigma_{old}|}\right)]$ and $C_\mu = \mathbb{E}_{\mu(s)}[\frac{1}{2}(\mu - \mu_{old})^T\Sigma^{-1}(\mu - \mu_{old})]$ is the split KL-constraint. We update both $Q(a, s)$ and $C(a, s)$ using RETRACE (Munos et al., 2016) which has empirically good performance and is guaranteed to converge under small assumptions.

Generally speaking, our E-step is written to allow for different factorizations, such as the one used in V-MPO (Song et al., 2019), which has higher per-update efficiency but comes at the cost of not being able to use off-policy data. For the sake of this work, we utilize the traditional MPO estimator, but V-MPO style updates should be a plug-in replacement.

---

**Algorithm 1** SafeMPO update step

---

**Require:** Policy $\pi$, safety function $K$, replay buffer $\mathcal{D}$, dual multipliers $\alpha_1, \alpha_2$, maximal dual $M_\lambda$
**Ensure:** $\pi > 0$
  sample Batch $B \sim \mathcal{D}$
  Update $Q(a, s)$ using RETRACE on $B$
  Update $C(a, s)$ using RETRACE on $B$
  solve Equation (10) for $\lambda \in [0, M_\lambda), \nu \in [0, \infty)$ on $B$
  Compute $q^\star(a|s)$ using Equation (11)
  **for** $0 \leq i < N$ **do**
     $\pi \leftarrow$ using Equation (12) descend on $\pi$
     $\alpha_1 \leftarrow$ using Equation (12) ascend on $\alpha_1$
     $\alpha_2 \leftarrow$ using Equation (12) ascend on $\alpha_2$
  **end for**

---

### 3.4 IMPLEMENTATION

Algorithm 1 shows the prototypical implementation of a single-batch update to our policy. We first fit both the Q and C function to the sampled trajectories using the RETRACE (Munos et al., 2016) estimator. We can then solve the dual problem Equation (10) and compute the nonparametric reference $q^\star(a|s)$, which we use as the target for our neural network. For the experiments in Section 4, we sample 20 batches of size 1024 and set $N = 8$.

## 4 EXPERIMENTS

We compare against state-of-the-art algorithms implemented in Omnisafe Ji et al. (2024). For all experiments, we choose $\kappa = 10$ and a budget of $B = 25$. Our benchmarks are performed on three tasks from the safety-gymnasium (Ji et al., 2023) benchmark set: **SafetyCarGoal** aims to drive a car in a randomly generated 3d environment towards a "goal" position while avoiding stationary hazards. Once a goal has been reached, a new goal is sampled. The reward is defined as the distance reduction towards the goal. The cost is $+1$ when an obstacle is hit, $0$ otherwise. **SafetyPointGoal** is the same task as the previous, but instead of navigating a differential drive car, the point agent directly decomposes rotation and movement direction, making control much easier. The agent is also physically smaller, further improving the navigation performance. **SafetyCarButton** is the most challenging environment: Instead of driving the car towards the goal and having to avoid stationary obstacles, the car has to avoid both stationary and *moving* obstacles, while navigating towards the correct goal button. Rewards and costs are defined the same as in the Goal environments.

Since CRL has two, often diametrically opposed objectives (constraint satisfaction and reward maximization), it is often difficult to compare whether one algorithm outperforms another. Further, one considerable issue when comparing different algorithms are the hypeparameters: This is of course a limitation for all ML algorithms, but in CRL we have the additional problem that the optimal solution to the CRL problem provably relies only on the Lagrangian parameter (Paternain et al., 2019). Therefore even a naive algorithm with fixed dual parameter could reach state-of-the-art when that parameter is known. This complicates benchmarking since over known benchmark environments practitioners know appropriate values for the dual, in contrast to real world problems where such ranges are unknown. This holds even for some dynamic methods, such as CPPOPID[5] (Stooke et al., 2020) which implicitly has environment information inside their PID-values.

To give existing methods the best possible opportunities, we utilize the highly optimized hyperparameters provided by Omnisafe (Ji et al., 2024). We would, however, argue that the fact that we only have a single hyperparameter which does not change the asymptotic behavior is significant advantage of our method that is hard to show in benchmarks.

When analyzing the results in Table 1, we observe that our method is competitive reaching state-of-the-art results in both the "SafetyCarGoal" and "SafetyCarButton" environments. Specifically,

---

[5]Note: we follow the nomenclature adopted by Omnisafe Ji et al. (2024) and call the combination of PPO (Schulman et al., 2017) and PID presented in (Stooke et al., 2020) "CPPOPID".

Table 1: Safety Car Goal. All methods are evaluated after 10 Mio steps. Following (Ji et al., 2024), we set the cost budget $B = 25$ for all algorithms unless otherwise mentioned.

| Method | SafetyCarGoal | | SafetyCarButton | | SafetyPointGoal | |
|--------|--------|------|--------|------|--------|------|
| | Reward | Cost | Reward | Cost | Reward | Cost |
| RCPO Tessler et al. (2018) | $18.71 \pm 2.72$ | $23.10 \pm 12.57$ | $-2.04 \pm 2.98$ | $43.48 \pm 31.52$ | $15.27 \pm 4.05$ | $30.56 \pm 19.15$ |
| PCPO Narasimhan (2020) | $21.56 \pm 2.87$ | $38.42 \pm 8.36$ | $0.36 \pm 0.85$ | $40.52 \pm 21.25$ | $18.57 \pm 1.71$ | $22.98 \pm 6.56$ |
| FOCOPS Zhang et al. (2020) | $15.23 \pm 10.76$ | $31.66 \pm 93.51$ | $0.21 \pm 2.27$ | $31.78 \pm 47.03$ | $17.97 \pm 9.01$ | $33.72 \pm 42.24$ |
| CPO Achiam et al. (2017) | $25.52 \pm 2.65$ | $43.32 \pm 14.35$ | $0.82 \pm 1.60$ | $37.86 \pm 27.41$ | $20.46 \pm 1.38$ | $28.84 \pm 7.76$ |
| CPPOPID Stooke et al. (2020) | $10.60 \pm 2.51$ | $30.66 \pm 7.53$ | $-1.36 \pm 0.68$ | $14.62 \pm 9.40$ | $8.43 \pm 3.43$ | $25.74 \pm 7.83$ |
| CUP Yang et al. (2022) | $6.14 \pm 6.97$ | $36.12 \pm 89.56$ | $1.49 \pm 2.84$ | $103.24 \pm 123.12$ | $14.42 \pm 6.74$ | $19.02 \pm 20.08$ |
| SafeMPO (ours) | $21.43 \pm 5.10$ | $32.23 \pm 7.43$ | $0.67 \pm 0.61$ | $30.87 \pm 4.47$ | $13.09 \pm 2.79$ | $32.92 \pm 3.43$ |
| SafeMPO $@B = 20$ (ours) | $20.24 \pm 2.24$ | $25.68 \pm 2.97$ | $0.94 \pm 0.12$ | $33.37 \pm 2.46$ | $18.36 \pm 3.41$ | $23.00 \pm 1.56$ |

(a) SafetyCarGoal      (b) SafetyCarButton      (c) SafetyPointGoal

Figure 2: Results throughout training. We adopt the same evaluation procedures as described in (Ji et al., 2024). The shaded area corresponds to $1\sigma$ deviations across independent runs. To maintain readability, we only plot a subset of the algorithms we compare against in Table 1.

our agent manages to reach a higher reward than CUP (Yang et al., 2022) and CPPOPID (Paternain et al., 2019), while producing substantially lower cost. The only method that produces a policy below the cost threshhold of 25 is RCPO (Tessler et al., 2018), which also substantially drops in observed reward compared to SafeMPO. The major downside of RCPO compared to our method becomes apparent in the more challenging SafetyCarButton environment. SafeMPO produces a slightly lower constraint violation than FOCOPS (Zhang et al., 2020) while providing almost $3\times$ more reward. The only method to stay fully within the constraint set (CPPOPID) does so at the loss of performance, producing a highly negative reward. RCPO - the best model from SafetyCarGoal - produces the worst results on SafetyCarButton: RCPO has the lowest reward and the second highest cost across all methods.

The pattern of inconsistent performance continues to SafetyPoinGoal - the easiest of the environments - here we find the best performing model to be CPO (Achiam et al., 2017) or PCPO (Narasimhan, 2020), depending on how one values a $\pm 3$ cost overrun/undershoot against a reward gain of $\approx 6$ points. This is the worst performing benchmark for SafeMPO, but we remain close to SOTA. Generally, our method has the advantage of behaving consistently, providing low run-to-run variance across independent runs.

Another interesting observation is the robustness provided by our method regarding costs: Our method consistently has the lowest run-to-run variance among all other approaches, which we hypothesize is due to the lack of "edge cases" introduced by other methods.

## 5 DISCUSSION

One interesting point to mention is that all our methods have a cost of roughly 30 compared to the cost limit of 25. This suggests that this is a limitation of how our likelihood function for safety is defined: since we only penalize policies *outside* the feasible set, the solution to Eq. 9 might end

up simply "eating" the cost for states that are almost safe already, leading to an effective shift in boundary due to sampling variance inside the approximations of the expectation. To validate our hypothesis, we re-run the SafetyCarGoal experiments with an artifically lowered cost threshhold of $B = 20$. The results can be found in table 1. As one can see, both SafetyPointGoal and SafetyCarGoal return feasible policies at SOTA reward levels (for feasible policies). Nevertheless, our method fails to find a feasible policy for SafetyCarButton, which is not unexpected considering that no method aside from CPPOPID can produce a feasible policy[6]. Of course, tuning the boundary $B$ is not a feasible solution but a smaller barrier leading to superior results suggests that our method is able to theoretically hit better threshholds, but is unable to reach the points due to the log-likelihood $G$ we chose. Our method can technically handle arbitrary notions of safety, including e.g. Conditional Value at Risk (CVaR). Since this problem goes deeper than SafeMPO and touches the question of "What definition of safety is appropriate for Reinforcement Learning", we leave the analysis of alternative safety likelihood functions for future work.

One important aspect to note is the general behavior of Reinforcement Learning in Constrained scenarios and the "RL as inference" notion employed in this paper. It is generally impossible for RL agents - regardless of Value, policy, or model based - to be 100% safe in unknown environments. This is due to the inherent uncertainty when estimating the constraints: To choose an extreme example, one cannot know the safety of states that are never sampled. This means any well designed CRL method should have the property that action probabilities are nonzero (but perhaps very low) during finite exploration, and only converge to zero in the limit of infinite data. Otherwise, one cannot guarantee that an action is "banned" just due to poor initialization of the cost function or missing data. Indeed, this is the behavior we observe in our proof of Theorem 4: High cost states decrease in likelihood but only hit zero in the limit of infinite steps where one has certainty on the cost function.

## 6    CONCLUSIONS AND FUTURE WORK

We propose a novel framework for Constrained Reinforcement Learning that relies on relative improvements in the likelihood of a policy being safe, rather than primal-dual or projection based updates. We first analyze these types of relative improvement algorithms theoretically, proving that naive improvement bounds may not lead to a contraction towards a feasible set. However, we prove that with small changes inspired by interior point barriers, we can make broad statements of the convergence properties under relative improvement constraints. Finally, we demonstrate our method's high performance by comparing against highly tuned state-of-the-art algorithms.

Our method leaves wide room for future work both on novel Safety Order Preserving Functions, and algorithmic improvements. For instance, we currently only consider the off-policy MPO algorithm (Abdolmaleki et al., 2018), but our framework is also compatible with other updates, such as (Song et al., 2019). As already mentioned in Section 4, we also noticed opportunities to improve our method further, by choosing a better likelihood function. Another interesting possibility is framing ordinary MDP RL problems as CMDPs where the objective is given as an improvement constraint. This way one could design RL methods that by construction guarantee improvement.

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

## A   LLM USAGE

We used LLMs to improve the writing and presentation of the work, as well as to aid in translation. We manually ensured that any alteration in wording did not change the meaning of the work.

## B   PROOFS

*Proof for Theorem 1.* Constructing the Lagrangian yields

$$\mathcal{L}(q, \eta, \lambda, \nu) = \mathbb{E}_{(a,s) \sim q}[Q(a,s)] + \eta(\varepsilon - KL(q\|\pi)) + \lambda K(q, \pi) + \nu \left(1 - \int q(a,s) da ds\right) \quad (13)$$

for $\eta, \lambda \geq 0$. Solving for the optimal $q$ (after absorbing the normalisation constraint) yields

$$q^\star(a,s) = \frac{1}{Z} \pi(a,s) \exp\left(\frac{Q(a,s) + \lambda \frac{\delta K(q,\pi)}{\delta q(a,s)}}{\eta}\right) \quad (14)$$

with $Z = \int \pi(a,s) \exp\left(\frac{Q(a,s) + \lambda \frac{\delta K(q,\pi)}{\delta q(a,s)}}{\eta}\right) da ds$ and $\frac{\delta K(q,\pi)}{\delta q(a,s)}$ the functional derivative wrt $q$.

Notice that for the Karush-Kuhn-Tucker conditions to hold, we also need complementary slackness of $\eta, \lambda$. Specifically we know that either $K(q, \pi) = 0$ or $\lambda = 0$.

In the former case, we have that the solution $q$ is no safer than the solution $\pi$, be definition. In the latter case, the solution reduces to

$$q^\star(a,s) = \frac{1}{Z} \pi(a,s) \exp\left(\frac{Q(a,s)}{\eta}\right) \quad (15)$$

which is the optimal solution to the unconstrained model. □

*Proof for theorem 2.* If $G(a,s)$ is not almost everywhere constant, then this means that there exists a pair of disjoint sets $X_1 \times X_2$ with $0 < \mu(X_1) = \mu(X_2) < \infty$ such that for all $((a_i, s_i), (a_j, s_j)) \in X_1 \times X_2$ we have $G(a_i, s_i) > G(a_j, s_j)$.

Choose $q$ such that $q(a_i, s_i) + \varepsilon_1 = \pi(a_i, s_i)$ and $q(a_j, s_j) - \varepsilon_1 = \pi(a_j, s_j)$. For any

$$0 < \varepsilon_1 < \min\left(\pi(a_i, s_i), \frac{1}{2}\sqrt{\frac{\varepsilon}{\mu(X_1 \cup X_2)}}\right)$$

Keep all other $q(a_k, s_k) = \pi(a_k, s_k) \forall k \neq i, k \neq j$.

The KL-constraint is upheld due to Pinsker's inequality, i.e for any $q, \pi$ we have

$$2D_{TV}(q, \pi)^2 \leq KL(q\|\pi) \leq \varepsilon.$$

Our distributions only differ on the set $X_1 \cup X_2$, meaning the total variational distance is less than $\varepsilon$ when we choose $\varepsilon_1 < \frac{1}{2}\sqrt{\frac{\varepsilon}{\mu(X_1 \cup X_2)}}$.

Trivially, the equality constraint $\int q(a,s)dads = 1$ is valid ($\varepsilon_1$ deviations cancel over pairs). The improvement constraint is nonzero:

$$\int_{\mathcal{A}\times\mathcal{S}} \pi(a,s)G(a,s)dads$$

$$= \int_{X_1} \pi(a,s)G(a,s)dads$$

$$+ \int_{X_2} \pi(a,s)G(a,s)dads + \int_{(\mathcal{A}\times\mathcal{S})\backslash(X_1\cup X_2)} \pi(a,s)G(a,s)dads$$

$$= \int_{X_1} \pi(a,s)G(a,s)dads$$

$$+ \int_{X_2} \pi(a,s)G(a,s)dads + \int_{(\mathcal{A}\times\mathcal{S})\backslash(X_1\cup X_2)} q(a,s)G(a,s)dads$$

$$< \int_{X_1} (\pi(a,s)+\varepsilon_1)G(a,s)dads$$

$$+ \int_{X_2} (\pi(a,s)-\varepsilon_1)G(a,s)dads + \int_{(\mathcal{A}\times\mathcal{S})\backslash(X_1\cup X_2)} q(a,s)G(a,s)dads$$

$$= \int_{X_1} q(a,s)G(a,s)dads$$

$$+ \int_{X_2} q(a,s)G(a,s)dads + \int_{(\mathcal{A}\times\mathcal{S})\backslash(X_1\cup X_2)} q(a,s)G(a,s)dads$$

$$= \int_{\mathcal{A}\times\mathcal{S}} q(a,s)G(a,s)dads$$

where we use the fact that values $G(a,s)$ in $X_1$ are strictly better than values $G(a,s)$ in $X_2$, and the fact that the sets $X_1$ and $X_2$ are not of measure zero to ensure strict inequality. Since $K$ preserves the strict inequality, $\int_{\mathcal{A}\times\mathcal{S}} \pi(a,s)G(a,s)dads < \int_{\mathcal{A}\times\mathcal{S}} q(a,s)G(a,s)dads \implies K(q,\pi) > 0$.

Therefore there exists a $q$ with nonzero $x$, which means $\log(x)$ is finite, which means a feasible interior point exists. The problem is also convex as both the KL constraint and the $K$ constraint are convex (assumption). This also directly implies Slater's condition, which implies strong duality. $\square$

*Proof for Theorem 3.* Consider the Lagrangian

$$\mathcal{L}(q,x,\lambda,\nu,\mu) = \int q(a,s)Q(a,s)dads + \kappa \log\left(\frac{x}{\kappa}\right)$$

$$+ \lambda\left(\int q(a,s)G(a,s)dads - \int \pi(a,s)G(a,s)dads - x\right)$$

$$+ \nu\left(\varepsilon - \int q(a,s)\log(\frac{q(a,s)}{\pi(a,s)})dads\right) + \mu\left(\int q(a,s)dads - 1\right)$$

The optimal $x$ is given by

$$x = \frac{\kappa}{\lambda} \tag{16}$$

while the optimal $q$ (after absorbing the normalization constant) is given by

$$q(a,s) = \frac{1}{Z}\pi(a,s)\exp\left(\frac{Q(a,s)+\lambda G(a,s)}{\nu}\right) \tag{17}$$

If $G$ is not constant eq. (9) has a solution, which can be found by optimizing over the dual variables $\lambda, \nu$. This follows directly from theorem 2.

If $G$ is constant, but $\lambda$ is bounded, the $q$ produced by optimizing eq. (9) can have any finite $\lambda$. However, since $G$ is constant $\lambda G(a, s)$ can be absorbed into the normalizer

$$
\begin{aligned}
q(a, s) &= \frac{1}{Z} \pi(a, s) \exp \left( \frac{Q(a, s) + \lambda G(a, s)}{\nu} \right) \\
&= \frac{1}{Z} \pi(a, s) \exp \left( \frac{Q(a, s)}{\nu} \right) \exp \left( \frac{\lambda G(a, s)}{\nu} \right) \\
&= \frac{1}{Z_2} \pi(a, s) \exp \left( \frac{Q(a, s)}{\nu} \right)
\end{aligned}
$$

which is the solution to MPO $\hspace{3cm}$ $\square$

## C    Convergence

*Proof of theorem 4.* To simplify the notation, we will define $P(q) = \mathbb{E}_{(a,s)\sim}[Q(a, s)]$.

We will show that the sequence $\tilde{q} = q_0, q_1, \ldots$ defined by recursively applying eq. (9) leads to a contraction towards the feasible set. First, consider the slightly simplified problem

$$
q_{i+1}^{\star} = \operatorname{argmax} P(q_{i+1}) + \kappa \log \left( \frac{K(q_{i+1}, q_i^{\star})}{\kappa} \right) \tag{18}
$$

Note that this corresponds to eq. (9) without the KL-divergence constraint.

Let's pick some reference point $q'$ within the feasible set. This acts as a lower bound for the maximization in eq. (18). By definition, we have that for

$$
P(q_{i+1}^{\star}) + \kappa \log \left( \frac{K(q_{i+1}^{\star}, q_i^{\star})}{\kappa} \right) \geq P(q_{i+1}') + \kappa \log \left( \frac{K(q_{i+1}', q_i^{\star})}{\kappa} \right)
$$

rearranging yields

$$
\kappa \log \left( \frac{K(q_{i+1}^{\star}, q_i^{\star})}{\kappa} \right) \geq [P(q_{i+1}') - P(q_{i+1}^{\star})] + \kappa \log \left( \frac{K(q_{i+1}', q_i^{\star})}{\kappa} \right)
$$

Due to boundedness of $P(q_{i+1}^{\star}) < M$ and finiteness of $P(q_{i+1}')$ we get

$$
P(q_{i+1}') - P(q_{i+1}^{\star}) \geq P(q_{i+1}') - M =: -\Delta_{i+1}
$$

This give us the rate

$$
\log \left( \frac{K(q_{i+1}^{\star}, q_i^{\star})}{\kappa} \right) \geq \frac{-\Delta_{i+1}}{\kappa} + \log \left( \frac{K(q_{i+1}', q_i^{\star})}{\kappa} \right) = \log \left( \frac{K(q_{i+1}', q_i^{\star})}{\kappa} \exp \left( \frac{-\Delta_{i+1}}{\kappa} \right) \right)
$$

which yields the inequality

$$
K(q_{i+1}^{\star}, q_i^{\star}) \geq K(q_{i+1}', q_i^{\star}) \exp \left( \frac{-\Delta_{i+1}}{\kappa} \right)
$$

One possible choice for $q'$ is any feasible point within the original CMDP $\mathbb{E}_{(a,s)\sim q'}[C(a, s)] \leq B$. Since this is feasible in every iteration, we get a fixed constant $c = \exp(-(M - P(q'))/\kappa) \in (0, 1]$ such that

$$
K(q_{i+1}^{\star}, q_i^{\star}) \geq c K(q', q_i^{\star})
$$

which is valid for all iterations. Therefore, we have a real decrease in $K(q_{i+1}^{\star}, q_i^{\star})$ in every iteration. By definition of Equation (18), we have a lower bound on $K$, which means $\tilde{q}$ converges.

We now have to show that it converges to a safe $q$ in the limit. For this consider

$$
K(q_{i+1}^{\star}, q_i^{\star}) \geq c K(q', q_i^{\star}) > 0
$$

Assume that we converge to an unsafe $q$, we then have a fixed point where $q = q_i^{\star} = q_{i+1}^{\star}$ via sequential continuity we have

$$
\underbrace{K(q, q) = 0}_{\text{def. } K} \wedge \underbrace{c K(q', q) > 0}_{\text{nonzero improvement}}
$$

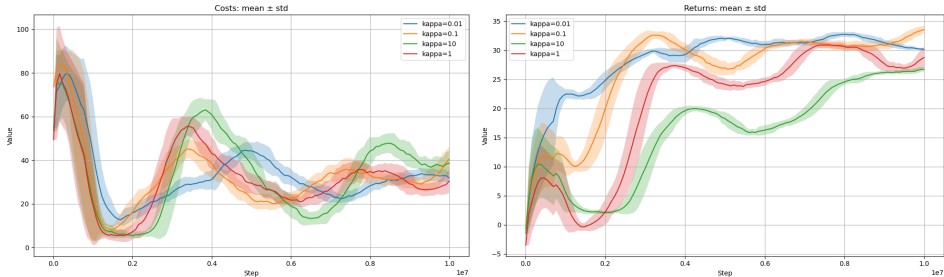

Figure 3: A comparison between different $\kappa$ values.

which is a contradiction. Therefore, we converge to a safe point in the limit.

Adding the KL-divergence bound complicates analysis, but the idea is the same. Picking a strictly feasible $q$ wrt the original CMDP constraints is not enough, since that might be outside the KL-ball. The fix is simple: Let

$$q'_{i+1} = \text{proj}_{KL(\cdot\|q_i)\leq\varepsilon}(q')$$

be the projection of the selected point onto the KL ball. Assuming $\varepsilon > 0$ we can perform exactly the same proof as above, just that instead of having a fixed rate $c \in (0, 1]$, we have

$$c_i = \exp\left(-\frac{M - P(q'_{i+1})}{\kappa}\right) \in (0, 1]$$

This still gives us an (iteration dependent) lower bound

$$K(q^\star_{i+1}, q^\star_i) \geq cK(q', q^\star_i)$$

where we can apply the same argument for convergence.

Now assume $K$ is the LSF. Let's call $c_{min} = \inf_i\{c_i\}$, we get geometric convergence via a standard squeeze argument:

$$K(q^\star_{i+1}, q^\star_i) \geq c_{min}K(q', q^\star_i) > 0$$

Meaning we have

$$K(q^\star_{i+1}, q^\star_i) \geq c_{min}K(q', q^\star_i)$$

for every step. Since the LSF telescopes $K(a, b) + K(b, c) = \mathbb{E}_a[G] - \mathbb{E}_b[G] + \mathbb{E}_b[G] - \mathbb{E}_c[G] = K(a, c)$, we have

$$K(q', q^\star_{i+1}) = K(q', q^\star_i) - K(q^\star_{i+1}, q^\star_i) \leq K(q', q^\star_i) - c_{min}K(q', q^\star_i) = (1 - c_{min})K(q', q^\star_i)$$

By induction we get

$$K(q', q^\star_n) \leq (1 - c_{min})^n K(q', q^\star_0)$$

which proves the geometric rate. $\qquad\square$

**Remark 3.** *Note that other choices for $K$ may equally lead to a geometric convergence rate, but for the sake of conciseness we only cover the convergence of the LSF used in our experiments.*

## D  ABLATION OF HYPERPARAMETERS

We run an ablation on the impact of $\kappa$ on the overall performance. We generally observed similar performance between the different $\kappa$ values across the entire interval tested.

We found no statistically significant difference in changing $\varepsilon$ slightly (evaluated across 10 runs).

# E  VALUE AND COST ESTIMATION

We use the RETRACE (Munos et al., 2016) algorithm to estimate the returns of $Q^\pi$ and $C^\pi$. RE-TRACE works by computing importance-weighted updates of the $Q$ function(s) over a replay buffer. Specifically

$$Q_{new}(s,a) = Q(s,a) + \mathbb{E}_\mu[\sum_{t \geq 0} \gamma^t (\prod_{i=1}^{t} c_i)(r_t + \gamma \mathbb{E}_\pi[Q(s_{t+1}, \cdot)] - Q(s_t, a_t),$$

where $\mu$ is an arbitrary behaviour policy (e.g. a previous version of the policy) and $\pi$ is the current policy. RETRACE can be implemented analogously for $C(s,a)$. RETRACE sets

$$c_i = \min(1, \frac{\pi(a_s|s_i)}{\mu(a_i|s_i)})$$

RETRACE is designed to give high quality estimates of the current policy by exploiting the similarity of previous policies to get lower variance estimates.

# F  HYPERPARAMETERS

We use a nearly identical network layout for policy, cost, and Q-function with 3 MLP layers with leakyReLU activations, followed by either a mean and covariance prediction layer (for the policy) or a value prediction head for cost/value networks. We use a width of 256 for all latent dimensions.

For the policy network we parameterize the mean and the cholesky factors of the covariance matrix.

We use $\kappa = 10$ and a standard $\varepsilon = 0.1$

# G  ENVIRONMENTS

Safety Gymnasium uses a "composable environment" format where every environment contains an "Agent" (in our case "Point" and "Car") and a task (in our case "Goal" and "Button").

The "Goal" tasks give as an input a 48 dimensional state which corresponds to the relative distance towards obstacles and the goal. The "Button" task consists of a 64 dimensional state which corresponds to the distance towards the goal, the buttons, the static obstacles, and the moving obstacles.

In addition to the task-intrinsic state, every agent also adds its own state, which for point is an accelerometer, velocimeter, gyroscope, and magnetometer (12 dimensions), and for the car are Quaternions of the rear wheel which are turned into 3x3 rotation matrices, Angle velocity of the rear wheel, accelerometer, velocimeter, gyroscope, magnetometer (24 dimensions).

Both "Car" and "Point" supply a 2 dimensional action space corresponding to a differential drive platform.

