# OpenReview forum: "SafeMPO: Constrained Reinforcement Learning with Probabilistic Incremental Improvement"
_ICLR.cc/2026/Conference — ICLR 2026 Poster_

### Official Review · Reviewer_4cCG · 2025-10-19

**Soundness:** 2
**Presentation:** 2
**Contribution:** 3
**Rating:** 2
**Confidence:** 4

**Summary:**

This paper presents a novel approach to constrained reinforcement learning, referred to as SafeMPO: Instead of relying on projection based approaches, which project the policy onto the constraint-satisfying set at every step, SafeMPO relies on incremental improvements by ensuring that each policy update monotonically increases the likelihood of safety, rather than enforcing hard constraint satisfaction at every step.

The motivation for this is that projection based approaches can be overly restrictive, effectively preventing the discovery of optimal, constraint satisfying solutions by enforcing strict constraint satisfaction at every step.

The paper builds on MPO and the EM algorithm, but extends MPO to handle additional safety constraints. In the E step, the method computes the improved, nonparametric distribution (in terms of return **and** constraint violation) $q^\ast$ over states and actions in closed form, subject to a KL constraint for $q^\ast$ to remain close to the current policy. Then, in the M step, the policy is fitted to the improved distribution $q^\ast$.

The paper provides theoretical guarantees showing that the incremental improvement process forms a contraction toward the feasible set, yielding geometric convergence under mild assumptions.
SafeMPO is evaluated empirically on three environments from the OmniSafe benchmark and compared with a number of appropriate baselines.

**Strengths:**

The proposed incremental approach to constrained reinforcement learning is, to the best of my knowledge, novel and well-motivated. Rather than enforcing strict constraint satisfaction at every update via projection of the policy onto the feasible set, the method ensures that each policy is at least as safe as the previous one. In settings where incremental progress toward the feasible set is acceptable, this formulation is intuitively less restrictive and may enable the discovery of higher-return, constraint-satisfying solutions that projection-based methods could fail to discover.

The proposed SafeMPO method is principled, and the paper's theoretical analysis provides guarantees for a practical constrained RL algorithm, including monotonic safety improvement and geometric convergence toward the feasible set under mild assumptions.

Finally, by leveraging a nonparametric surrogate optimization in the E-step and a supervised M-step, the method inherits the stability and low-variance optimization behavior of MPO-style algorithms while extending them naturally to constrained settings.

**Weaknesses:**

The proposed approach is principled and I very much agree with the motivation and would like to see (an improved version of) this work published. However, the following points strongly impact my score:

**(1) Clarity and exposition**
While the paper uses typical notation from variational inference, the presentation of key background material should be improved substantially. The paper assumes strong familiarity with MPO and the EM algorithm, but neither is introduced sufficiently, making the paper not self-contained.

In particular, the introduction of MPO (lines 120–124) is confusing and somewhat misleading. What exactly is
$q$ here? Unlike what is stated, and if I am not mistaken, $q$ is not an approximation of the RL objective, but rather an auxiliary, implicit distribution over states and actions used to optimize it.

Nitpick: There are also a few other statements that lack clarity, IMO. For example, the feasible set is never properly defined. The $\epsilon$-ball on line 67 is not defined. Referring to “the neural network” as part of SafeMPO is problematic, since there are multiple neural networks in RETRACE alone (RETRACE is also mentioned only briefly and not explained in a self-contained manner).

**(2) Experimental validation**
The empirical results are limited and somewhat contradictory. In all reported environments, the standard SafeMPO variant fails to find a policy satisfying the cost constraint (B=25), as the asymptotic costs after 10M steps remain above this threshold. The found policies are therefor **not** part of the feasible set. The chosen environments are relatively simple navigation tasks, making this result somewhat concerning. Lowering the cost bound (to B=20) to achieve feasibility (on 2/3 environments) is not an adequate solution, as it effectively alters the optimization objective in a way similar to tuning the relative weights of reward and cost. Thus, the experiments do not convincingly demonstrate that the proposed algorithm converges to constraint-satisfying solutions in practice.

**(3) Disconnect between motivation and findings**
The authors argue that the incremental approach in SafeMPO is less restrictive and can therefore discover solutions that are hard to find using projection-based methods, as nicely illustrated in the didactic example in Figure 1. It would strengthen the paper’s claims considerably if the authors demonstrated an environment where such exploration challenges due to constraints exist, and where SafeMPO successfully overcomes them.

**(4) Lack of environment and implementational details**
The environments are not sufficiently explained. The state and action spaces are not stated, making it hard to know how easy or hard these environments truly are. Similarly, the hyperparameters of SafeMPO aren't stated, limiting the interpretability and reproducibility of the results.

**Questions:**

1. In equation 8a, the term $\kappa\ \log (\frac{x}{\kappa})$ seems to be constant in (s,a). How does it affect maximization over $Q(s,a)$?

2. Why do all almost all of your baselines fail to find feasible solutions, even on the simplest environment? Might it be that there is insufficient capacity in the estimators for the cost function or policy networks?

3. Why does the performance of SafeMPO in terms of return increase on 2/3 environments when a stricter cost coefficient is use?

---

> ### Author Response · Authors · 2025-11-24
>
> We first and foremost want to thank the reviewer for reading our work and highlighting our contributions to the theoretical analysis of our method.
> Regarding your concerns
> > While the paper uses typical notation from variational inference, the presentation of key background material should be improved substantially
>
> We generally follow the structure used in prior work extending MPO, such as V-MPO. We do understand that MPO is a nontrivial algorithm to understand, but we are also somewhat limited due to the finite number of pages available to us: It is simply beyond the scope of a conference paper to introduce foundational concepts in variational inference.
> Nevertheless, we can try to address your concerns by updating our manuscript:
> For the sake of understanding the prior work of MPO, really the only relevant things one needs to understand are equations 2 and 3, where one first optimizes a surrogate distribution $q$ (equation 2), which then gets “cloned” into the neural network by minimizing the error between the policy network $\pi$ and the proposed solution $q$ (this is equation 3).
> Basically, while MPO has a really complex derivation, the only things that matter for our work is the resulting algorithm, which is “Sample state-action pairs for equation 2, solve equation 2 for $q$, clone $q$ into the policy network using equation 3”.
> Do you have any specific things you think are missing to further increase your understanding?
>
> $q$ is just the solution of the optimization problem just like in $\min_x f(x)$ the variable $x$ is defined as the solution of the optimization problem.
> There is an unfortunate naming-clash between the variational inference community which calls the variational distribution $q$ (lowercase) and the RL community which calls the state-action function $Q$ (uppercase). We decided to choose this nomenclature as it lines up with the naming used in the original MPO paper and minimizes the confusion for people that have a background in variational inference and reinforcement learning.
> The feasible set is defined as the set of policies that are within the constraints (equation 1), and the $\varepsilon$-Ball is defined in line 67. We have updated our work to clarify this. We have also changed all mentions of “the neural network” to “the policy neural network” to make it clear that we refer to the policy network and not e.g. Q or C networks.
>
> We have also added a brief explanation of the RETRACE algorithm in the appendix. Note that for our algorithm, the exact details of RETRACE do not matter: RETRACE is simply a well-established way to estimate the Q and C targets.

---

> > ### Author Response · Authors · 2025-11-24
> >
> > >Experimental validation
> >
> > I think it makes sense to tackle this in combination with your question
> > >  Why do all almost all of your baselines fail to find feasible solutions, even on the simplest environment? Might it be that there is insufficient capacity in the estimators for the cost function or policy networks?
> >
> > and
> > >Disconnect between motivation and findings
> >
> >
> > In short, solving CMDPs is still an open problem and there is currently **no method** that can reliably solve given CMDPs. In fact, it is still open whether some benchmarks, like SafetyCarButton even _have_ safe solutions which can be found by policy based RL (note that the one method within the constraints finds an exploit in the environment, hence why it is safe, but also has negative reward). This was the result of the large “omnisafe” benchmark paper, which is where we base all our experiments on. Constrained RL is deceptively difficult due to the interaction between exploration (trying to maximize state coverage) and constraints (trying to "delete" parts of the state space). Constraints add an "unknown unknown" to the environments which make exploration in constrained RL exceedingly difficult. In general, the environments used in omnisafe considered are incredibly complex for current CRL methods.
> >
> > Our method is theoretically guaranteed to find a feasible policy in the limit of infinite exploration just like other unconstrained algorithms. This is not strictly true for other CRL methods, since they often only guarantee that they can stay within the feasible region once they have entered it. The reason our method is unable to find a feasible solution for SafetyCarGoal and SafetyPointGoal is due to sampling noise. In practice, we don’t evaluate $\mathbb{E}\_q[G(a,s)]$, but only take $|batchsize|$ many samples to approximate the objective. This means that the likelihood we get is now $\mathbb{E}\_q[G(a,s)]\pm error$.
> >
> > However, since our method is consistent (you can see this at the best-in-class run-to-run variance in the cost terms), we can simply move the likelihood by “error” and get back a feasible policy (basically setting $B=20$ causes SafeMPO to “see” a $B=25$ boundary since sampling error is added to the estimate).
> > Neither our method, nor the baselines perfectly solve CMDPs since solving CMDPs is currently out-of-reach in general. Our method is interesting because it offers good properties from a theoretical point of view and, aside from the steady-state error introduced by sampling, is able to produce safe policies in all cases where a safe policy is known to exist.
> >
> > However, we do agree that all CMDP methods are still fundamentally imperfect: Our method is a step on the path towards true safety, but surely not the final word from an algorithmic point of view.
> >
> > >Why does the performance of SafeMPO in terms of return increase on 2/3 environments when a stricter cost coefficient is use?
> >
> > This is because more policies are (implicitly) excluded from the search. There is nothing fundamentally that “forces” constrained RL to be harder than unconstrained RL. In constrained RL you just have fewer options, which can be a blessing or a curse. Our method is special in how it handles the "curse" (i.e. exploration in constrained spaces) part, so it makes sense that it performs well under tightened restrictions.
> >
> > >Lack of environment and implementational details
> >
> > We use the standard benchmarks as outlined in the safety-gymnasium (and omnisafe) paper(s). We have added a brief explanation of this and of the hyperparameters to the appendix.
> > Can you have a look whether that is sufficient for you?
> >
> > We hope to have cleared your questions, but please reach out to us again, specifically regarding the details of the MPO algorithm. We of course cannot reproduce the entire MPO paper as part of the introduction, but It would help us a lot if you could tell us what specifically you think would be good to add to the explanation!

---

> > > ### Comment · Reviewer_4cCG · 2025-11-25
> > >
> > > Thank you for the response and the additional clarifications.
> > >
> > > I still find the paper fairly difficult to follow, but I am not factoring this in very strongly. The explanations provided by the authors as part of the rebuttal are pedagogically helpful, and I encourage the authors to incorporate some of these into the main paper, especially given that there is still space available. However, there is no code provided as supplementary material, and the authors have not committed to releasing it for the camera-ready version, which negatively affects the reproducibility of the work.
> > >
> > > I unfortunately remain unconvinced by the experimental results, which I find underwhelming and not fully aligned with the motivation stated in the introduction. In particular, SafeMPO does not appear to demonstrate clear performance gains over more restrictive baselines as a result of its incremental safety approach. Concerns about the strength of the empirical results was also raised by Reviewers y7UB and MFV7. The rebuttal’s argument that “currently no method can reliably solve the given CMDPs” does not not reassure me fully, especially since the OmniSafe benchmarks indicate that several existing algorithms perform well on CMDPs very similar to those evaluated in this paper.
> > >
> > > That said, I do believe the paper presents a promising and potentially valuable approach to solving CMDPs. In light of this, I am increasing my score to 4. I do see limitations in the paper, but none so severe that the paper couldn't be published.

---

> > > > ### Author Response · Authors · 2025-11-25
> > > >
> > > > First, thank you for raising your score and your very quick response to our rebuttal!
> > > >
> > > > > However, there is no code provided as supplementary material, and the authors have not committed to releasing it for the camera-ready version, which negatively affects the reproducibility of the work
> > > >
> > > > This was our mistake: We will of course release the code after acceptance. For now, we have added an annonymized version of our code to the supplemental material. After acceptance, we will do a proper release via github.
> > > >
> > > > > The rebuttal’s argument that “currently no method can reliably solve the given CMDPs” does not not reassure me fully, especially since the OmniSafe benchmarks indicate that several existing algorithms perform well on CMDPs
> > > >
> > > > To clarify our previous statement: There are methods which can solve _individual_ CMDPs, but not CMDPs in general. For instance, if we randomly pick the first CMDP method shown in the omnisafe paper (table b on page 11), which is RCPO, you can see that the method performs really well on SafetyCarGoal (reward +18 with costs $23.10\pm 12.57$), but really badly on SafetyCarButton (Reward -2, costs $43.48\pm 31.52$) and SafetyPointGoal (reward 15 and costs $30.56\pm19.15$). The same thing is true with other methods, like CUP. Our method has high run-to-run consistency (i.e. no $\pm 31.52$ swings), and is rather consistent between the benchmark environments.
> > > >
> > > > However, we do not want to argue that SafeMPO is a gigantic jump in capabilities beyond the state-of-the-art. The interesting part of our work is the theoretical analysis of a bayesian view on incremental feasibility improvements, not the top-line performance.
> > > >
> > > > We once again want to thank the reviewer for their time and quick response and we will do our best to improve the introduction of MPO in the background section.

---

### Official Review · Reviewer_MFV7 · 2025-10-27

**Soundness:** 3
**Presentation:** 2
**Contribution:** 3
**Rating:** 4
**Confidence:** 3

**Summary:**

This paper presents SafeMPO, a constrained reinforcement learning algorithm derived from the maximum a posteriori policy optimization (MPO) framework. The method aims to guarantee asymptotic safety constraint satisfaction with contiguous policy improvement. The authors formulate an expecation maximiation algorithm with a constrained optimization problem in the E-step (Theorem 1), introduces a modified policy update that includes a safety barrier (Eq. 8a), and provides theoretical results relating the algorithms contraction and monotonicity properties to safety performance. Empirically, SafeMPO is evaluated on standard continuous-control safety benchmarks (SafetyGym) and compared to existing constrained policy optimization baselines.

**Strengths:**

- I find the idea interesting and novel. Combining MPO-style policy updates with constrained optimization is a natural direction that should provide some interesting insights. The attempt to incorporate constraint satisfaction directly into the policy updates of an MPO-style algorithm is conceptually nice.
- The derivation of the constrained optimization E-step are motivated optimization. The authors include a variety of additional theoretical results (contraction and convergence) to their algorithmic contribution.
- The motivation for combining the probabilistic policy optimization framework with constrained objectives is clear, and the paper structure (E-step / M-step / theory / experiments) is easy to follow.
- The paper is overall well-written.

**Weaknesses:**

**Conceptual and theoretical clarity**:

- I found the E-step section, particularly around Theorem 1, quite hard to follow. It is not clear to me why an optimal solution to Theorem 1 would not guarantee a safety improvement, as constraint (7c) seems to directly enforce this. Any solution violating safety improvement should be infeasible under that constraint, so the subsequent introduction of the barrier function in Eq. (8a) is hard to comprehend from my position. Going to the appendix reveals that the authors likely mean that the first order KKT conditions permit points that are not feasible, but these conditions in my understanding are not a characterization of global optima. Either way I believe this should be explained more explicitly, as it lays the foundation for the motivation of subsequent algorithmic components.
- The barrier function added as an additional regularizer in (8a) makes the algorithm less "pleasing" to me. If I am correct, the introduced term behaves similarly to a Lagrange multiplier or penalty term, which raises the question of whether the same problem is being solved twice - once in the primal constraints and again in the added regularizer. In general I find it slighly confusing that the authors seem to pursue a "a nonzero improvement in our safety constraints" but do not formalize their primary constraints as such, instead permitting zero-improvement (7c).
- The authors’ usage of the term "contraction mapping" (Corollary 2 and 3) confuses me a bit. Contractions are properties of operators, yet the text seems to apply the notion to entire inequalities. It is a bit unclear to me what mapping is claimed to be contractive, and with respect to which norm or metric.

**Conceptual tension with the control-as-inference view**:

This is a discussion on a more philisophical level, but I believe there is a tension between soft control formulations (where policies emerge from defining optimality from a lens of energy-based exponential distributions) and hard safety constraints. The paper incorporates constraints onto a probabilistic inference objective, but what does the introduction of hard constraints imply for the soft nature of the probabilistic treatment of control? In the classical control-as-inference setting, maximum entropy solutions naturally emerge as solutions to the probabilistic problem, which appear to be incompatible with hard constraints if left untreated. This conceptual issue perhaps deserves some discussion.

**Experiments**:

The experiments look executed well but their results look rather inconclusive to me. The differences between SafeMPO and existing constrained methods are minor across most benchmarks, and it is unclear in which scenarios SafeMPO’s particular constraint handling makes a significant difference. The authors should better articulate where SafeMPO is expected to excel (e.g., in nonconvex or highly stochastic safety settings) and provide corresponding experiments.

**Questions:**

- Could you clarify in why the optimal solution to Theorem 1 does not guarantee safety improvement? Does this follow from a characterization through KKT conditions? If so, do these condition allow a general statement as done about the global optimal solutions?
- How does the behavior of the barrier function in Eq. (8a) differ from a standard Lagrange or penalty term? Why is the non-zero improvement of constraint satisfaction permitted in the primary constraints in Eq. (8a) if it appears to be the authors wish to ensure non-zero improvement?
- In what sense is the E-step mapping a "contraction"? What operator or function is being iterated, and in which norm?

---

> ### Author Response · Authors · 2025-11-24
>
> We first and foremost want to thank the reviewer for reading our work and especially their focus on the theoretical results.
> Regarding your outlined weaknesses:
> > It is not clear to me why an optimal solution to Theorem 1 would not guarantee a safety improvement, as constraint (7c) seems to directly enforce this
>
> This point exactly hits at the subtle issue in the optimization problem 7:
>
> Notice that 7c does **not** guarantee improvement in the constraints, only that the results will not get any worse than they are currently!
> This is exactly the reason we do not have guaranteed movement towards the feasible set, which is what we prove in theorem 1. The only scenario where eventual feasibility is guaranteed is if $ \mathbb{E}\_{(a,s)\sim q}[Q(a,s)]$ is always higher in feasible regions than in infeasible regions.
>
> However, in this special case all constraint are vacuously true: Just running MPO would give exactly the same results since the constraints don’t matter if following greedily maximizing $Q$ never violates a constraint.
> Even if you solve eq 7 to a global optimum, the result will not, in general, be safer: Consider, for instance, the linear safety function, we can rewrite this as
> $$\mathbb{E}\_q[G(a,s)] - \mathbb{E}\_\pi[G(a,s)] \geq 0 \iff \mathbb{E}\_q[G(a,s)] \geq \mathbb{E}\_\pi[G(a,s)]$$
>
> If we are in the “equals” case $\mathbb{E}\_q[G(a,s)] = \mathbb{E}\_\pi[G(a,s)]$, we do not improve in safety. Theorem 1 shows that you are always in the equals case, unless your $Q$ is defined in a way where feasible points are always better than infeasible (i.e. unconstrained RL solves the problems).
> So, either equation 7 doesn’t improve (in which case it is not a guaranteed to be feasible in the limit), or it degenerates to an MPO update (in which case it is not a CRL method at all).
> In short constraint 7c is not an “improvement constraint” but rather a “don’t get any worse” constraint.
>
> Did this resolve your question?
> >The barrier function added as an additional regularizer in (8a) makes the algorithm less "pleasing" to me
>
> The logarithm trick fixes this since we cannot have a point directly on the boundary. This is usually an issue in interior-point methods (since they have a hard time finding values at the boundary) but this is a feature for us since we precisely do not want to have the boundary $\mathbb{E}\_q[G(a,s)] = \mathbb{E}\_\pi[G(a,s)]$. The reason we formulate our constraints as inequalities, rather than strict inequalities has to do with optimization theory.
>
> In optimization, one rarely considers inequalities since they break existence of optimal points. Say your optimization problem is simply $\min x\ s.t.\ x>1$, you no longer have a well defined optimum: Is the optimum $1.1$, or $1.01$ or $1.001$? In general, one can only define the supremum/infimum of these optimization problems since, formally, you can be arbitrarily close to the boundary (and this is before you consider the numerical issues of implementing $x>y$ in floating points). As an aside: In the limit of infinite optimization steps ''$>$'' and ''$\geq$'' give the same results, so even in the theoretical “pure math” sense just forcing $x>y$ does not work.
>
> We sidestep that issue since we don’t want an “as small as possible” improvement, but rather a “sufficiently large” (i.e. nonzero) improvement: We don’t care whether it is the minimum possible improvement, a medium sized improvement, or a large improvement.
> The challenge we have to solve is that we want an arbitrarily nonzero improvement exactly _how_ large it is doesn’t matter for us.
>
> Did this explain your questions?
>
> >The authors’ usage of the term "contraction mapping" (Corollary 2 and 3) confuses me a bit
>
> We treat the optimization problem (7 and 8 respectively) as a map from $\pi\to q$.
> This is the same sense of how it is used for Value iteration or for e.g. proximal operators. We altered the wording in the corollaries a little bit, can you check whether this helps?
> The contraction “norm” we consider is a contraction wrt the distance to the feasible set, as measured by $K(\cdot,\cdot)$. I.e. if you have a feasible policy (e.g. the optimal $\pi^\star$), the “distance” as measured by $K$ decreases. Note that we do not need $K$ to be a norm, it is sufficient for it to be a safety order preserving function.
> Do you think we should include an explicit explanation for “optimization results as operators from $\pi\to q$”?

---

> > ### Author Response · Authors · 2025-11-24
> >
> > >Conceptual tension with the control-as-inference view
> >
> > We completely agree with you, and, in fact, would go even further: All “classical” formulations of RL work poorly in the context of constraints.
> > Fundamentally, any approach (be that inference based or classical value iteration) has the fundamental tension between learning and acting safely:
> > It is generally impossible for RL agents - regardless of Value, policy, or model based - to be 100\% safe in unknown environments. This is due to the inherent uncertainty when estimating the constraints: To choose an extreme example, one cannot know the safety of states that are never sampled. This means any well designed CRL method should have the property that action probabilities are nonzero (but perhaps very low) during finite exploration, and only converge to zero in the limit of infinite data. Otherwise, one cannot guarantee that an action is ``banned'' just due to poor initialization of the cost function or missing data. Indeed, this is the behavior we observe in our proof of Convergence: High cost actions decrease in likelihood but only hit zero in the limit of infinite steps where one has certainty on the cost function.
> > One can technically also deal with this issue outside of control-as-inference, but the inference view has a really good notion of the information gathered from the environment (this is the reason KL-divergence bounds appear naturally).
> > The interesting part of our formulation specifically is the flexibility in your notion of safety, as discussed in the response to 3c3x, one can change the safety likelihood function to capture different notions of safety.
> > However, the way one should model constraints is – in our opinion – an open problem in general which goes way beyond the current CMDP settings.
> > We have added a discussion of control-as-inference for CRL into the manuscript, though it is worth noting that this tension exists in all CRL methods, the design of control-as-inference methods just makes it readily apparent (which is not necessarily a bad thing!)
> >
> > Do you think this is a sufficient discussion of control-as-inference?
> >
> > >it is unclear in which scenarios SafeMPO’s particular constraint handling makes a significant difference
> >
> > It is quite hard to design benchmarks around this since our method specifically excels if the environment is complex enough that a feasible start becomes difficult to realize. This is also why we focus our testing on navigation tasks where the initial e.g. “just drive forward” solution is generally infeasible. You can see some of the advantage of being able to handle such infeasible starts in the low run-to-run variance in table 1.: Since our method works well, even if the initialization is poor, we have much lower cost variance as we don’t get bogged down in local optima. However, such things are hard to benchmark conclusively.
> > Our work is generally more focused on building the theoretical framework around CRL with incremental constraints and Bayesian inference. We do not claim that this method is as-is a state-of-the-art CRL method. The important part for now is that the method works and is theoretically well founded.

---

> > > ### Comment · Reviewer_MFV7 · 2025-11-26
> > > **Response to Authors**
> > >
> > > Thank you for the detailed response. I found them insightful in general:
> > >
> > > 1. *Safety improvement in 7*: Thank you, this does clarify the issue mainly for me. Perhaps a follow-up to be sure: If you made an additional assumption that one has access to a safe initial policy, the solution to the problem formulation in 7 should guarantee safe policies throughout the learning process, correct?
> > > 2. *Barrier functions in 8*: The response clarifies the issue for me. The non-zero "step-size" of the improvement could lead to interesting behaviors in practice from what I can see. Did you observe signifcant differences in behaviors ( e.g. in exploration) depending on $\kappa$?
> > > 3. *Contraction maps*: I see. Still, in value iteration I am used to having notation of explicitly defined operators (e.g., the Bellman operator). I indeed would find it useful to clarify, even as a brief note in in one place, how the optimization problem or its solution constitutes an operator and that $K(\cdot, \cdot)$ constitutes the distance in which this contraction is measured.
> > > 4. *Conceptual tension with the control-as-inference view*: I appreciate the author's added discussion text. I tend to agree that a tension between safety and RL persists in a very general sense.  What I was getting at, at a superficial level, is that the notion of defining a "probability of a trajectory being optimal", which is inherent to the control as inference perspective, leaves some space for different interpretations in constrained RL problems. E.g., I believe the authors define a joint probability of a trajectory being both safe and optimal which remains soft in nature. I was wondering if there were cosntructions more "faithful" to the intuition behind constrained RL, e.g., by baking the *hardness* of the safety of trajectories into the optimality of a trajectory, e.g., as $P_\pi(O=1) = 0 if \sum_t c_t > C$ (and as usual otherwise). I do not know if this makes for a sensible algorithm, but it struck me as a point for discussion and I do maintain that I find the approach in the paper sensible as well.
> > >
> > > I believe the authors provided good responses and changes and I am inclined to raise my score if the authors address the small remaining points above.

---

> > > > ### Author Response · Authors · 2025-11-27
> > > >
> > > > Thank you for your swift response! We are happy we could alleviate some of your questions and we think we can solve your additional questions as well:
> > > > > If you made an additional assumption that one has access to a safe initial policy, the solution to the problem formulation in 7 should guarantee safe policies throughout the learning process, correct?
> > > >
> > > > Not necessarily: You also need to guarantee that $C^\pi(a,s)$ is good enough. Specifically, if your learned cost model $\hat C(a,s)$ is a proper overestimator of the true costs $\hat C(a,s) \geq C(a,s)$ then formulation 7 guarantees that a safe policy remains safe (assuming we can evaluate the expectations exactly). However, the downside of this is obvious: If your $\hat C(a,s)$ happens to be large at a point where the true $C(a,s)$ is not, then that state-action pair will never be tried by $\pi$ and therefore will never be corrected to the lower value!
> > > >
> > > > > The non-zero "step-size" of the improvement could lead to interesting behaviors in practice from what I can see. Did you observe signifcant differences in behaviors ( e.g. in exploration) depending on $\kappa$?
> > > >
> > > > We did not observe any significant differences in how $\kappa$ is set (we have added a plot into the appendix). In theory, we would expect a lower $\kappa$ to lead to more exploration and slower convergence towards the feasible set, while a larger $\kappa$ does the opposite. It is very likely that for any reasonable choice of $\kappa$ the differences are so marginal that the effect is lost in the natural sampling noise present in all real-world RL processes.
> > > >
> > > > > Contraction maps: I see. Still, in value iteration I am used to having notation of explicitly defined operators (e.g., the Bellman operator).
> > > >
> > > > No problem, adding a small note to address this is quite easy!
> > > > We will add this in the next revision once the other reviewers have issued their feedback.
> > > >
> > > > > . I was wondering if there were constructions more "faithful" to the intuition behind constrained RL, e.g., by baking the hardness of the safety of trajectories into the optimality of a trajectory, e.g., as $P_\pi(O=1) = 0 if \sum_t c_t > C$ (and as usual otherwise).
> > > >
> > > > This was our initial idea as well, but this quickly produces issues like that the feasible region within the KL-Ball might be empty (or, without the KL-ball: the step is larger than is justified by the gathered data), or that $\sum c_t$/$C(a,s)$ is noise/wrong. It is quite easy to construct scenarios where such a constraint problem fails due to the policy/constraint-estimator being in a bad position.
> > > > The way you would tackle this issue of $P_\pi(O=1) = 0 if \sum_t c_t > C$ being noisy/dependent on unknown quantities, is by turning the indicator into a random variable (with some likelihood function, this is our $G(a,s)$). Once you turn the indicator into a random variable, you now need to define with what likelihood you are going to allow an infeasible trajectory to be accepted (which is going to be more than 0% of the time due to uncertainty). In our case, we defined this threshold to be “you should accept an infeasible trajectory less often than you did in the last iteration”, which gives rise to the $K(\cdot,\cdot)$ framing.
> > > > It is very possible that there are different ways to frame this problem, but as long as you don’t know the constraints ahead of time you have to have some way of dealing with the uncertainty introduced by you not knowing where the constraints are placed.
> > > >
> > > > We want to once again thank the reviewer for their time and would be very appreciative if the score were to be raised

---

### Official Review · Reviewer_3C3X · 2025-10-31

**Soundness:** 3
**Presentation:** 3
**Contribution:** 3
**Rating:** 6
**Confidence:** 3

**Summary:**

This paper proposes SafeMPO, a novel algorithm for Constrained Reinforcement Learning (CRL) that extends Maximum a Posteriori Policy Optimization  to the CMDP setting. Instead of using primal–dual updates or projection-based recovery, SafeMPO enforces *incremental safety improvement* via a *log-barrier constraint on a safety-order-preserving function \(K\)*. The method yields a closed-form E-step distribution \(q^*(a|s)\) and theoretical guarantees that the iterates contract toward the feasible set under mild assumptions. Empirical results on **Safety-Gymnasium** tasks demonstrate competitive or superior reward-cost trade-offs compared to strong baselines like CPO, PCPO, RCPO, and CPPOPID, with notably low variance across runs.

**Strengths:**

- The paper offers a novel probabilistic viewpoint on safe RL: treating safety improvement as a likelihood maximization problem rather than a constraint projection.
 - The barrier-regularized E-step and its EM-style update are well derived and integrate naturally with existing MPO framework.
- Theoretical contributions are well-structured. The paper first proves that naive improvement bounds do not necessarily contract toward feasibility, then introduces a log-barrier to guarantee convergence (Theorems 2–4). I did not check all the theoretical results but they look correct and rigorous
- The algorithm looks simple.
- Experiments across three standard safety environments show SafeMPO achieving near-SOTA returns with moderate constraint violations.

**Weaknesses:**

- **Experimental scope is limited.**
  Only three tasks from the Safety-Gym benchmark are evaluated, which is insufficient to demonstrate the generality of the approach. The study would be strengthened by including a broader range of tasks to better assess scalability and robustness.

- **Ablation studies are underdeveloped.**
  The authors should conduct sensitivity analyses on key hyperparameters—specifically the step-size parameter \(\kappa\), the KL divergence radius \(\varepsilon\), and the maximal dual variable \(M_\lambda\)—since the theoretical convergence guarantees explicitly depend on these quantities.

- **No exploration of alternative safety likelihoods.**
  Although the paper acknowledges that other likelihood formulations beyond the truncated exponential are theoretically possible, no experiments evaluate such alternatives. Empirical validation of different safety likelihoods would be necessary to justify the exclusive use of the truncated exponential form.

- **Baselines are outdated.**
  The experimental comparison omits several recent state-of-the-art constrained RL algorithms, such as **Constrained Variational Policy Optimization (CVPO)** [1] and imitation-based constrained methods [2]. Including these baselines is essential for establishing the competitiveness of the proposed approach.

**Refs**

- [1] Liu, Z., Cen, Z., Isenbaev, V., Liu, W., Wu, S., Li, B., & Zhao, D. (2022). *Constrained Variational Policy Optimization for Safe Reinforcement Learning*. In *Proceedings of the 39th International Conference on Machine Learning (ICML 2022)*, PMLR 162, 13644-13668. [https://proceedings.mlr.press/v162/liu22b.html](https://proceedings.mlr.press/v162/liu22b.html)

 - [2] Hoang, H., Mai, T., & Varakantham, P. (2024). Imitate the Good and Avoid the Bad: An Incremental Approach to Safe Reinforcement Learning. In Proceedings of the 38th Annual AAAI Conference on Artificial Intelligence (Vol. 38, No. 11, pp. 12439-12447). AAAI.

**Questions:**

Please address the points I raised in the Weakness.

**Details Of Ethics Concerns:**

I do not identify any major ethical concerns with this work. The study focuses on algorithmic development and evaluation in simulated safety-critical reinforcement learning environments, without the use of real-world human or animal data. However, as the proposed method targets safety in autonomous systems, future deployment should consider ethical implications related to reliability, accountability, and potential misuse in high-stakes domains.

---

> ### Author Response · Authors · 2025-11-24
>
> We first and foremost want to thank you for reading our paper and highlighting the strengths of our probabilistic and theoretically well-founded work.
> Regarding your questions:
> > Experimental scope is limited
>
> Safety-Gymnasium is designed around a “composable” set of environments which combine an agent (e.g. car or point) with a task (e.g. goal or button). This means that one has an exponential number of different environments which, while technically different, are substantially similar. For this reason we opted to instead use a statistically significant number of independent runs for every method. At the end of the day, we are limited by computational resources (especially since we consider a significant number of baselines), so we have to balance the number of environments against how many runs we can perform.
> We selected the environments based on the environments which were nontrivial in the omnisafe benchmark.
>
> > Ablation studies are underdeveloped
>
> We have added a parameter sweep over the $\kappa$ parameter into the appendix.
> We are waiting on the results for $\varepsilon$, but we do not expect significant differences since $\varepsilon=0.1$, is a default value in PPO, MPO, TRPO, etc…
> The maximal dual variable $M_\lambda$ is simply any large number. Basically, it is just for mathematical cleanliness, practically $M_\lambda$ has no effect other than that we have to stop optimizing at some point.
>
> In general, **none** of the above mentioned change whether our algorithm converges or doesn’t. It just trades off speed against robustness (in the case of $\varepsilon$) or whether the algorithm preferentially pushes down the cost or up the reward (in the case of $\kappa$). In the limit, both variants will be identical and converge to the same optima.
>
> Did this clarify your concerns?
>
> >No exploration of alternative safety likelihoods
>
> We agree that this is a really interesting avenue for future research. Fundamentally, constrained reinforcement learning still has big open questions, such as “What does it mean to be safe”, which is exactly what the safety likelihood defines.
> The current safety likelihood function is for the classical CMDP setting where one has to keep the expected costs bounded $\mathbb{E}[C(a,s)] \leq B$. The current likelihood function follows the canonical choice for mapping the costs to a distribution, just as $\exp(R(\tau))$ is the canonical choice for returns (see MPO).
>
> Changing the safety likelihood function changes what it “means” to be a safe policy. For instance, Conditional Value at Risk metrics penalize the long-tails of a cost function. Our method is well equipped to study these different notions of safety since we keep the safety likelihood flexible, but a thorough exploration of the difference between these likelihoods goes far beyond the scope of this paper, since it also has to answer what it means to be a safe policy.
> We have added the above to a new “discussion” section, and also added the comment on why the currently chosen safety likelihood is canonical to the manuscript.
>
> > Baselines are outdated
>
> The reason we chose the baselines we did is because they were reproducibly high performers in the “omnisafe” benchmark suite. The issues with many prior works on CRL is that benchmarks were scarcely standardized and different papers referred to different benchmarks by the same name.
> For instance, CVPO uses a fundamentally different variant of the environments we consider: From the CVPO paper
>
> >Particularly, we increase the simulation time-step and decrease the timeout steps for each environment, such that the agent can finish the tasks with fewer steps. In addition, the Goal task in this paper is modified from the Button task, since we can then fix the layout of the goal buttons and obstacles to make the environment more deterministic. Note that the original SafetyGym implementation will random sample the layout for each episode, which greatly increase the training time and variance [...]
>
> CVPO fixing the layout of the constraints fundamentally changes the problem, since now “absolute” positions can be learned, while agents in the randomized setting can only react to a change in layout dynamically. Basically, the benchmark in CVPO allowed the RL algorithm to overfit to a specific scenario.
>
> This is a persistent issue in CRL, so we rely on the independent reproduction and standardization of algorithms and benchmarks provided through omnisafe.
> In omnisafe, all algorithms where evaluated and tuned to a consistent standard, which makes it appealing for reproducibility purposes. In general, omnisafe algorithms tend to perform better than in their original implementation, which is due to extensive tuning and implementation optimization.
>
> Do you think there is any fundamental issue with this approach?
>
> We hope we were able to address all your concerns (aside from the $\varepsilon$ results, which we will deliver as soon as they are ready).
> Do you have any additional questions?

---

### Official Review · Reviewer_y7UB · 2025-10-31

**Soundness:** 3
**Presentation:** 3
**Contribution:** 3
**Rating:** 4
**Confidence:** 3

**Summary:**

This paper introduces a constrained policy optimization algorithm based on Maximum-a-Posteriori Policy Optimization (MPO), providing theoretical guarantees for convergence to the feasible set of the constrained problem. The approach extends MPO’s control-as-inference formulation by adding a “policy is safe” event alongside the “policy achieves goal” event. The authors argue this improves conditioning, as safety can be improved even outside feasibility. Experiments in Safety-Gymnasium (Ji et al., 2023) show partial empirical gains.

**Strengths:**

- Novel idea of redefining MPO using a safety probability event
- Engaging and well-motivated introduction
- Theoretical results appear sound (not fully verified)

**Weaknesses:**

- Experimental results are not fully convincing
- Expected stronger sample efficiency gains since MPO is off-policy
- Theoretical clarity: Corollary 3 seems central but is not well explained
- Mathematical presentation issues: $ C(a,s) $ is estimated but never defined or used; unclear whether this refers to $ G(s,a) $ or $ K(s,a) $. Clarify how these are estimated via Retrace.

**Questions:**

- Why can the most likely posterior policy not be non-feasible?

---

> ### Author Response · Authors · 2025-11-24
>
> We first and foremost want to thank  the reviewer for their efforts in reviewing our work. We especially want to thank  them for highlighting the strengths of our work (Extending MPO to a safety event and a strong theoretical backbone).
>
> Regarding your highlighted weaknesses:
> Generally, all CMDP solvers have significant weaknesses in their application across different Environments if benchmarked to the rigorous standards laid out in Omnisafe. This can be clearly seen in table 1, where only two methods (Ours and CPPOPID) even manage approximately satisfying the constraints in more than one benchmarking problem (and CPPOPID only does this by finding an exploit in the SafetyCarButton environment).
>
> This is also why we do not discuss sample efficiency in the context of this paper: We do agree that – in theory – an off-policy method should outperform an on-policy method regarding sample efficiency, but this is a bit of a mute discussion if solving CMDPs is still not reliable.
>
> In short: Sample efficiency was not a focus of this work because solving CMDPs is a problem even without considering sample complexity.
>
> Regarding Corollary 3: This immediately follows from theorem 2. Theorem 2 proves that the optimization problem 8 has a solution (i.e. finite objective and q) and strong duality holds. If we know a solution exists, we also know that, by design $x>0$ (since otherwise 8a would not be finite). If $x>0$ this means that $K(q,\pi)>0$ which then means that
>
> $$\mathbb{E}\_{(a,s)\sim q}[G(a,s)]>\mathbb{E}\_{(a,s)\sim \pi}[G(a,s)]$$
>
>  (by construction of definition 1). Which means that the new policy $q$ is safer than the old policy $\pi$. This means that if we repeatedly apply eq 8, we will slowly move towards the feasible set ($\implies$ we contract towards the feasible set).
> In short: As long as 8 has a solution, it has a solution with $x>0$, which means we improve in terms of safety. If we apply 8 repeatedly, we slowly move towards the feasible set (i.e. optimization 8 is a contraction mapping $\pi\to q$).
>
> Did this clarify your question or are there still open problems?
> > Mathematical presentation issues: is estimated but never defined or used
>
> Thank you for your comment: We did indeed only define $C(\pi)$. For the sake of this work, we define $C(s,a)$ as the cost associated with executing action $a$ in state $s$ and following the policy after (analogous to the Q-function). We have added this to the document. The $C(a,s)$ is used to define the safety likelihood function in equation 5 (we have also altered the presentation of equations 5 and 6 to make this more clear).
>
> Did this clarify the meaning of $C(a,s)$?
>
> Regarding your question
> > Why can the most likely posterior policy not be non-feasible?
>
> I’m not sure what exactly this question is referring to, but if it refers to our comment in line 189-192, then the reason is rather straightforward:
> Imagine you have the option between a policy with reward 100 but excess costs of 2, and the policy with reward -100 but 0 excess costs (for the sake of this toy example, assume $\alpha=\beta=1$ weighting). In this case, the log-likelihood in equation 4c would be  100+exp(-2) for the first option, and -100+1 for the second option.
>
> This means that in this case the log-likelihood of the unsafe policy is much higher than the safe one.
> The way you prevent this is by scaling up the weighting $\beta$ of $p(S=1|\tau)$ until the cost-likelihood outweighs the reward likelihood. However, if you do this naively (e.g. set $\beta=\infty$), you have zero exploration (you bet that $p(S=1|\tau)$ is already a perfect estimator from the beginning) and you can easily violate your KL divergence.
> This is exactly the problem we discussed in the introduction.
>
> We prevent this by automatically choosing $\beta$ and $\alpha$ such that the policy doesn’t collapse (i.e. the KL-divergence is bounded like in MPO) while still guaranteeing that the costs reduce by a meaningful amount. The core difficulty in guaranteeing this is that we do not know anything about the dynamics of the environment, the distribution over rewards, the costs, or even whether the initial policy is feasible.
>
> Since our framework is bayesian, we can deal with estimation errors in the costs in a well defined manner: we update towards the feasible set only in so far as it is supported by the data.
>
> We hope we were able to address all issues raised and are open for any further questions!

---

> ### Comment · Reviewer_y7UB · 2025-11-27
>
> Thanks to the authors for clarifying and answering our questions.
>
> In particular:
> > We do agree that – in theory – an off-policy method should outperform an on-policy method regarding sample efficiency, but this is a bit of a mute discussion if solving CMDPs is still not reliable.
>
> This sounds reasonable. You have convinced me with that.
>
> The questions regarding feasibility of posterior policy has been answered as well. I would thus, upgrade my rating to 6.
>
>
> Thanks a lot.

---

### Author Response · Authors · 2025-11-28

Considering all the feedback we have received by the reviewers, we thought to summarize the current state of the discussion across all reviews in a general comment:

All reviewers noted the novelty of our Bayesian view of Constrained RL.

They also noted the novelty of our incremental safety guarantee.

All reviewers also commended the theoretical analysis of our method.

The main issues noted by reviewers were mostly connected with the perceived poor performance of our method.
For this we noted that all existing CRL methods appear to have low performance under the strict evaluation criteria put forth by omnisafe. We want to once again note that omnisafe benchmarks are not considered solved! Specifically, prior work had high run-to-run variance and high inter-environment variance that our method reduces significantly.

We also improved the wording in the Corollaries 2 and 3, and the introduction of the prior work on MPO to make the results clearer. We also added an ablation of the $\kappa$ parameter in the appendix, which shows that changing $\kappa$ has limited to no effect on the final model.

Finally, we added a discussion section to specifically talk about the general control-as-inference view from the perspective of constrained RL and the reasons why one might want to change the safety likelihood function.

We believe we addressed all major open points and are still open for additional discussion.

---

### Meta-Review · Area_Chair_gUd6 · 2025-12-27

**Summary:**

Across all four reviews, this paper is viewed as a reasonable and conceptually interesting extension of Maximum a Posteriori Optimization (MPO) to constrained reinforcement learning (RL) via a probabilistic (i.e., safety-likelihood) formulation, with a generally sound technical direction. After the discussion and rebuttal, the reviewers' consensus shifted to an accept-leaning majority (approximately 6, 6, 6, 4). The rebuttal successfully addressed key technical questions around feasibility interpretation, including why intermediate and posterior policies may be non-feasible, and clarified the motivation and benefits of the proposed approach relative to existing constrained-RL baselines. The remaining concerns primarily center around the empirical validation and reproducibility commitments.

Overall, given the strengthened consensus among reviewers and the rebuttal’s clarifications, I recommend acceptance. I would expect that the revised version will further tighten experimental reporting, add key robustness and ablation evidence, and potentially to provide a concrete reproducibility plan if possible.

**Reviewer Concerns:**

The authors' rebuttals successfully address the reviewers' concerns in terms of
- The rebuttal clarified how feasibility should be interpreted in the proposed probabilistic constraint framework. This includes why intermediate and posterior policies may violate constraints even when the procedure targets safer updates overall.
- Reviewers raised concerns about whether the proposed method’s benefits are consistent given data-collection and optimization differences across baselines. The rebuttal clarified the intended comparison and when SafeMPO is expected to be advantageous.
- In the rebuttal, the authors provided additional explanations around the algorithmic design choices such as how the "safety likelihood" is incorporated into the MPO-style updates and why this yields incremental improvement behavior. Those explanations help reduce confusion about what is novel relative to existing constrained RL methods.

The major concerns in the paper that were not fully addressed during the rebuttal center around the empirical evaluation.  The rebuttal and follow-up discussion indicate additional analyses and ablations were added or planned, one reviewer were unconvinced as those should be added during the discussions (but the discussion schedule were unexpected changed). In the revised version, the authors are expected to further tighten experimental reporting, add key robustness and ablation evidence, and potentially to provide a concrete reproducibility plan if possible.

**Reviewer Scores:**

Reviewer y7UB explicitly stated that they would upgrade their rating to 6 after the rebuttal addressed their main concerns.

Reviewer 3C3X did not explicit state score update, however, their main concerns have been partially resolved during the rebuttal. I would expect little to no change in the score.

Reviewer MFV7 acknowledged that the rebuttal was responsive and indicate they are inclined to raise the score if a few remaining points are addressed. Given the authors’ follow-up clarifications, I’d predict a score increase to 6.

Reviewer 4cCG explicitly updated their score upward to 4 after rebuttal, while still expressing concern about experimental results and reproducibility.

---

### Decision · Program_Chairs · 2026-01-26

Accept (Poster)